# Forty Years without Family: Three Novel Bacteriophages with High Similarity to SPP1 Reveal Decades of Evolutionary Stasis since the Isolation of Their Famous Relative

**DOI:** 10.3390/v14102106

**Published:** 2022-09-23

**Authors:** Véronique A. Delesalle, Brianne E. Tomko, Albert C. Vill, Katherine B. Lichty, Greg P. Krukonis

**Affiliations:** 1Department of Biology, Gettysburg College, 300 N Washington St., Gettysburg, PA 17325, USA; 2Department of Molecular Biology and Genetics, Cornell University, 526 Campus Rd., Ithaca, NY 14850, USA; 3Department of Biological Sciences, University of Delaware, Wolf Hall, Newark, DE 19716, USA; 4Department of Biology, Angelo State University, Cavness Science Building 101, ASU Station #10890, San Angelo, TX 76909, USA

**Keywords:** bacteriophage, phage, SPP1, *Bacillus subtilis*, comparative genomics, functional annotation, gene conservation

## Abstract

SPP1, an extensively studied bacteriophage of the Gram-positive *Bacillus subtilis*, is a model system for the study of phage–host interactions. Despite progress in the isolation and characterization of *Bacillus* phages, no previously fully sequenced phages have shared more than passing genetic similarity to SPP1. Here, we describe three virulent phages very similar to SPP1; SPP1 has greater than 80% nucleotide sequence identity and shares more that 85% of its protein coding genes with these phages. This is remarkable, given more than 40 years between the isolation of SPP1 and these phages. All three phages have somewhat larger genomes and more genes than SPP1. We identified a new putative gene in SPP1 based on a conserved sequence found in all phages. Gene conservation connotes purifying selection and is observed in structural genes and genes involved in DNA metabolism, but also in genes of unknown function, suggesting an important role in phage survival independent of the environment. Patterns of divergence point to genes or gene domains likely involved in adaptation to diverse hosts or different environments. Ultimately, comparative genomics of related phages provides insight into the long-term selective pressures that affect phage–bacteria interactions and alter phage genome content.

## 1. Introduction

The last two decades have seen a resurgence of interest in bacteriophages (phages) for many reasons. First, studies of the ubiquity of phages have led researchers to estimate the global population of phages at 10^31^, establishing them as the most numerous and ecologically important entities in our biosphere [1,2,3,4]. Second, the majority of phage genes have no known function, making phage genetics the “dark matter” of genetic diversity [5,6,7,8]. Third, phages impose strong selective pressures on bacteria as is evident from the numerous defense mechanisms of their hosts and the counter-defenses of these parasites [9,10,11,12,13,14,15]. In some cases, these interactions move from being antagonistic to mutualistic, as when phage genes are incorporated into bacterial genomes and provide their hosts with some fitness advantage [8,16,17]. Fourth, these interactions have significant impacts on microbial community structure and the processes that are dependent on these bacterial communities such as disease transmission, decomposition, and nutrient cycling [18,19,20,21]. Finally, as antibiotics become less effective at controlling pathogenic bacteria, phage therapy has become an increasingly attractive alternative [22,23]; whether phages are a viable substitute for antibiotics, however, requires a better understanding of the ecology and evolution of phages, in particular what conditions allow phages to move from one host to another.

Despite growing interest in the above topics, much of what we know about phage biology is dependent on a small cadre of phages (e.g., the T-phages, lambda, SPP1). SPP1 was isolated in 1968 from soil in the Botanical Garden of Pavia, Italy [24] and has been the subject of numerous structural studies e.g., [25,26,27,28]. It was sequenced in 1997 by [29] and resequenced in 2018 by [25]. In a 2014 study [30] that classified 83 *Bacillus* sequenced phages, SPP1 was recognized as not sharing genetic similarity with any isolated phages. That state of affairs has been changed by our work. Our lab is focused on the ecology and evolution of *Bacillus subtilis* phages, particularly from desert environments. As an initial step, we have isolated and sequenced over 30 *Bacillus* phages from soil collected across the Southwest United States. Among these, three phages stood out for their significant genetic similarity (>80%) to SPP1. In this paper, we explore the comparative genomics of these phages and SPP1. These comparisons have allowed us to identify a novel SPP1 gene and genes which may not be needed for life in a lab environment. In addition, our data suggest the possibility that in adapting to the lab environment SPP1 has experienced deletions that have led to a smaller genome and fewer genes compared to our “wild” phages.

## 2. Materials and Methods

### 2.1. Soil Collection and Phage Isolation

Soil samples were collected from various sites in the Southwest United States and specifically for this report around Tucson, Arizona (Table 1). At each location, soil was collected in a sterile 50 mL conical tube and GPS coordinates were recorded. To isolate phages from these samples, approximately 1 g of soil was added to 50 mL of modified Lysogeny broth (LB) containing 10 mM CaCl_2_, 10 mM MgCl_2_, and 1 mM MnCl_2_ (LB2). Soil cultures were incubated at both 30° and 37 °C while shaking at 250 rpm, and 2 mL aliquots were sampled from each temperature flask immediately after soil was added to the broth and thereafter at 1.5, 3, 6, 12, 24, and 36 h of shaking. Liquid samples were filtered (0.22 μm polyethersulfone) and stored in cryotubes at 4 °C.

From a subset of these processed samples, spot testing was used to ascertain the presence of phages. Following protocols described in the SEA-PHAGES Discovery Guide [31], for samples showing lysis, single plaques were purified and amplified to obtain high titer lysates. Working stocks of phage lysates were stored at 4 °C, and archives of isolated phages were stored in 40% glycerol at −80 °C. Isolated phages were named based on soil sample number, location, and phage isolate. For example, 049ML003 is the third phage isolated from soil sample site 049, collected on Mount Lemmon.

### 2.2. Bacterial Strains and Cultures

Phages were isolated on various *Bacillus subtilis* strains: the lab strain 168 obtained from the Bacillus Genetics Stock Center [32] (BGSCID 1A1) and “wild” strains originally obtained from Dr. Conrad Istock and now in the personal collection of Dr. Krukonis. These wild strains have been shown to be genetically and physiologically different from one another and from existing strains [33,34]. The phages described here were isolated on T89-06 or T89-19, both isolated from soil in the Sonoran desert (Table 1). Stock spore cultures were grown while shaking at 250 rpm and 37 °C for 72 h in media formulated to enhance sporulation [35]. Spore cultures were then washed by repeated centrifugation and resuspended in sterile distilled deionized water, and titers were determined by plating and spreading serial dilutions of these cultures on LB agar. From these cultures, spore stocks of 10^7^–10^8^ spores per mL were generated and kept at 4 °C.

### 2.3. Phage DNA Isolation, Sequencing, and Genome Assembly

For the newly isolated phages, DNA was extracted from high titer lysates (at least 10^9^ PFU/mL) using the Promega Wizard DNA Clean-Up System. DNA libraries were prepared and sequenced on an Illumina MiSeq platform by North Carolina State University’s Genomic Science Laboratory. Reads were aligned and assembled using GS de novo assembler (default settings; Roche Life Science) and verified in Consed [36,37]. Each genome assembled into one contig and consensus sequences were determined to be of high quality with no additional sequencing necessary. Genome ends were determined using Pile-Up Analysis for Starts and Ends [38] and/or PhageTerm [39]. When phages have circularly permuted genomes, by commonly accepted convention [37], we set the first base as the upstream start of the small terminase. All genome annotations have been submitted to NCBI (accession numbers in Table 1).

### 2.4. Annotation and Comparative Genomics

The finished sequences were imported into DNA Master [40] to map and compare open reading frames (ORFs). Putative genes were called based on both Glimmer and GeneMark algorithms [41,42]. Putative functions of gene products were predicted using protein homology search software, including BLASTp [43] and HHpred [44]. For BLASTp matches an E-value below 10^−5^ was necessary to consider assigning a function. For HHpred matches, a high probability (>85%), substantial coverage (>50%) and low E-value (<10^−5^) were necessary. For proteins with high probability and low E-values but low coverage, a conserved domain rather than a function could be assigned. In addition, gene order (synteny) was also used to predict gene function for structural and assembly genes [45]. The absence/presence of tRNA genes was determined by running genomes through the programs tRNAscan [46] and Aragorn [47]. Average Nucleotide Identity (ANI) and Average Amino acid Identity (AAI) were calculated using algorithms from [48]. Dotplots of genome nucleotide sequences were created with Gepard (at word length 10) [49]. To compare differences in the amino acid sequences of homologous genes, the relevant nucleotide sequences were imported into Mega7 [50], then the translated amino acid sequences were aligned using Muscle (default parameters), and the number of SNPs and indels for each pair of homologous genes determined. Alignment figures were made with ESPript [51]. Additional analyses were conducted using BLASTn (nr/nt database; done in July 2022) or BLASTp (nr database; done in July 2022) as appropriate, either against the complete database or the databases restricted to tailed phages (taxid: 10699, 10662, and 10744), using default parameters.

A Phamerator database [52] was constructed for our phages as well as other completely sequenced *Bacillus* phages. Our database included 250 phages: 33 *Bacillus* phages isolated in our lab, including the 3 phages that are the focus of this paper, and 217 completely sequenced phages available on NCBI as of 1 May 2019: 214 *Bacillus* phages and three *Geobacillus* phages (GBK2, GSV1, and TP-74) with sequence similarity to *Bacillus* phages. This database was used to generate genome maps that illustrate relationships between phages based on both amino acid and nucleotide similarity. Proteins are grouped into phage protein families, or phams, based on pair-wise comparisons if they exhibit BLASTp alignment with an E-value below 0.001 and a ClustalW alignment of greater than 32.5% amino acid similarity. Shared gene content as well as ANI data and dotplots were used to group phages into clusters, sets of phages that share nucleotide similarity over more than half of their genome [7,30,53,54]. Following the cut-off point recently used by [30,53,54], phages that shared 35% or more of their phams were classified as belonging to the same cluster (phages that share less than 35% of their phams with other phages would be classified as singletons).

## 3. Results

### 3.1. Three Phages Share High Nucleotide Similarity with SPP1 and Form a New Cluster

Three phages isolated around Tucson, Arizona (Table 1) are notable for their nucleotide and amino acid similarity to SPP1 (Figure 1). Despite being isolated more than 40 years apart, SPP1 and our phages share significant similarity (and at least 82.5% for nucleotide and 78.2% for amino acids; Figure 1) with 100% query coverage (based on BLASTn data). Bacillus phage 000TH010 was isolated in 2011 and has the largest genome among these phages with 90 putative ORFs. The other two phages, 049ML001 and 049ML003, were isolated from the same gram of soil collected in 2014. Despite this geographical closeness, these two phages only shared 98.6% ANI and 90.0% AAI with each other (Figure 1). All of these phages have bigger genomes with more genes than SPP1, especially 000TH010 (Figure 2; genome map for this phage). These three phages were isolated on wild strains of *B. subtilis* (Table 1) and all produce clear plaques on their isolation host, suggesting they are virulent phages, a conclusion supported by our annotations of their genomes. Although isolated on strains of *B. subtilis* different from 168, all our novel phages are capable of lysing *B. subtilis* 168 (Delesalle and Krukonis, personal observation). Sequence similarity to SPP1 clearly indicates that these phages belong to the Siphoviridae family, a conclusion supported by transmission electron microscopy (Figure 3; TEM for 000TH010). They also have circularly permuted genomes, whose orientation and start have been aligned to SPP1. As expected for phages of *Bacillus* species, they have a low GC content.

### 3.2. What Our SPP1-like Phages Reveal about SPP1

In a 2014 study that classified 83 *Bacillus* phages, SPP1 was recognized as a singleton [30]. Based on the criteria described above [30,53,54], our three phages and SPP1 form an original and distinct cluster from other sequences *Bacillus* phages. Using SPP1 as the query genome, a BLASTn search reveals three other phages with much more limited nucleotide similarity to the phages described here: phage Ray17 (13% query coverage and 80.5% identity; accession MH752385), PM1 (7% query coverage and 82.0% identity; accession AB711120), and vB_BspS_SplendidRed (9% query coverage and 80.2% identity; accession MN013088). 

We compared the genomes of our phages to that of SPP1, using the recently revised annotation of SPP1 [25]. Our genome sequencing and annotation supported the proposed changes to SPP1 genome (Table 1 in [25]) but also revealed interesting details and patterns (Table 2). First, we were able to call a function for three SPP1 genes based on HHPred matches or recently published studies [26] (in bold in Table 2). Second, we called one new putative gene of unknown function in SPP1 (gene 46.2; in italics in Table 2) based on similar sequences (putative homologs) in our phages. Although this gene is small (102–105 bp), the high nucleotide conservation for this gene and its flanking genes among these four phages support our call (see also Figure 4). The ability to identify novel genes even in well-studied phages is one of the benefits of having close relatives for genomic comparisons. This should be particularly true for small genes, which are less likely to be called by gene prediction programs. Third, our genomic analysis suggests that one pair of adjacent genes in SPP1 maps to a single gene in two of our phages. More specifically, between the head and scaffolding genes, genes 8 and 9 in SPP1 as well as genes 11 and 12 in 000TH010 correspond to one gene (gene 6) in both 049ML001 and 049ML003 (Figure 4). Although SPP1 gp 8 retains sufficient protein homology to the gene products of 049ML phages to be classified in the same pham, these proteins show significant differences, including numerous large in-frame indels (up to 56 amino acids; Figure 5). In addition, 000TH010 gp 11 is classified as belonging to a different pham, probably due to larger indels (Figure 4 and Figure 5). Moreover, although few genes are very different between the two 049ML phages, their gene 6 shows more genetic differences than is typical for genes between these two phages, particularly in the middle of the gene (Figure 4 and Figure 5). In contrast, there has been little divergence in the amino acid sequence for gp 9 in SPP1, gp 12 in 000TH010, and the corresponding homologous portion of gp 6 in 049ML001 and 049ML003 (Figure 4 and Figure 5). We cannot determine whether this is the case of one gene being split into two in SPP1 and 000TH010 by a fortuitous mutation that results in a stop codon before a start codon or the case of two genes being fused into one in the 049ML phages, nor can we determine whether these genes are ultimately processed as one or two peptide products without proteomic analysis. However, comparison of their amino acid sequences does suggest that selection is acting differently on these genes or gene domains among these phages. The high conservation of the homologs of SPP1 gp 9 suggests that this product is functional for all of these phages. Ref. [25] did not identify a function for this gene product and our analysis does not provide additional insight on this question.

### 3.3. Genome Conservation and Genes under Strong Purifying Selection

These four phages have a genome organization that is typically observed in Siphoviridae phages [7,45,55]: a structural cassette with strong gene synteny (terminase, small and large subunits—portal—scaffolding protein—major capsid protein—head–tail connectors—major tail protein—tail assembly chaperones—tape measure—minor tail proteins) followed by a lysis cassette (lysin and holin) and a replication cassette (a set of genes involved in DNA metabolism). In all genomes, all genes are on the “forward” strand. There is strong amino acid conservation (often 100% alignment and greater than 95% amino acid identity; Table 2) among most of the structural proteins but also among some gene products associated with DNA metabolism such as the RecT-like recombinase, suggesting strong purifying selection on these genes despite more than 40 years of viral evolution. In addition, some gene products of 000TH010 such as the tail tip protein (gp 21 in SPP1) or the holin components (gp 24.1) have greater amino acid similarity to their SPP1 homologs rather than the 049ML001/003 homologs. This is despite the fact that 000TH010 has the lowest ANI and AAI with the other three phages.

### 3.4. Genome Mosaicism

We identified a total of 116 phams in these phages. SPP1 had homologs in the other three phages for all but 12 gene products. In total, 57 phams were shared by all SPP1 phages, with 37 of these having an identified function (Figure 5). Moreover, 26 phams were found in more than one but not all of these four phages, and 33 phams were found in only one phage. Only one of these phams could be assigned a putative functional domain: the protein product of ORF 72 in 049ML001 has HHPred matches to a DNA-J-like Zn binding domain. Most of the unique phams are found in 000TH010. The unique phams are found in two locations: (1) in the right arm of these genomes for all phages, a region that thus shows less nucleotide conservation among phages; and (2) early in the genome, among structural genes, in 000TH010. As could be expected, the phams that appear novel fall into two categories when using BLASTp to search for homologs: gene products with no significant matches in the current NCBI database or with partial matches to hypothetical proteins in various Bacillales (i.e., *Bacillus*, *Brevibacillus*, *Geobacillus*, *Paenibacillus*). The latter matches suggest horizontal gene transfer, that these genes may have been picked by ancestral phages from their bacterial hosts.

### 3.5. Identifying Genes or Gene Domains under Different Selection Pressures

Despite the high nucleotide similarity among these phages, we observed regions with either insertions/deletions of small putative genes and regions of lower similarity within homologous genes. Here we bring attention to some of these regions (unless otherwise indicated gene numbers refer to genes in SPP1). First, between the genes for the large subunit of the teminase and the portal, we find a highly conserved gene of unknown function in all four phages (gene 4). In both SPP1 and 00TH010, this gene is flanked by unique small genes also of unknown function. No such flanking genes are found in 049ML001 and 049ML003 (Figure 5). Second, the capsid gene and protein show less nucleotide/amino acid conservation for the second half of their sequences when comparing SPP1 to our phages: we noted 15 versus 33 amino acid differences, respectively, for the first versus last 164 amino acids in this protein. Third, we observed strong amino acid and nucleotide conservation among the structural genes from the head-to-tail adaptor to the first holin, as well as for the lysin and the second holin genes (this region corresponds to genes 15 to 26). However, comparisons among these four phages, especially comparing 000TH010 to the other phages, showed two patterns: (a) regions of high nucleotide and amino acid conservation in the middle section of the tape measure gene (from aa 721 to 866), the distal tail, the first half of the tail tip, gene 24 of unknown function, and the first holin; and (b) genes with regions of low nucleotide similarity but still high amino acid conservation for the head-tail joining (gene 17), the major tails and tail assembly chaperones, the tape measure, the second half of the tail tip (gene 21 in SPP1), and the putative tail protein. These comparisons provide insights into how selection may be acting on these structural genes (see discussion)

Finally, gene 42.1 is a case where HHP red matches strongly suggest a DNA binding function for its gene product. However, the putative homolog in 000TH010 (gene 68) has sufficiently diverged that no such function can be ascribed to its gene product. Moreover, a one-bp insertion (a sixth A) in 049ML001 resulted in an early stop codon and a presumably functionless protein for this phage (i.e., a pseudogene; high sequencing coverage indicates that this is an insertion rather than a sequencing error). This appears to be a case of a gene under different selective pressures in different phages, although we do not know what the selective forces are.

## 4. Discussion

We have only begun to isolate and describe the estimated 10^31^ phages in the biosphere. We still know very little about global phage diversity [56] or even the diversity of phages that infect a particular host. Our knowledge of phages also lags far behind our knowledge of their hosts [57]. Currently the best-described phages are the Actinobacteriophages [58]. As of 1 August 2022, 4114 sequenced Actinobacteriophages [59] have been divided into 158 clusters, with new phage isolation and sequencing leading to the characterization of new clusters on a regular basis. In contrast, as of 1 August 2022, fewer than 500 *Bacillus* phages have been fully sequenced and entered into NCBI. It is thus expected that newly isolated *Bacillus* phages are more likely to be genetically different from previously isolated ones, although the diversity of isolation hosts and geographical locations sampled is likely to impact this expectation (isolating on different hosts is likely to isolate genetically distinct phages while isolating from the same location is less likely). The oldest isolated Actinobacteriophages are D9, D32, and L5, all isolated in 1954 [59]. They belong to cluster A2, which, as of 1 August 2022, contains 102 members. It was thus exciting to isolate for the first time, in more than 40 years, three phages with a very high level of genetic similarity to SPP1.

As detailed in our results, the isolation of SPP1 relatives provides insights into genome organization and evolution impossible when a phage has no close relative. First, our genomic comparisons have allowed us to identify a novel gene (gene 46.2). Given recent evidence for the expression of small genes in phages [60,61], we suggest that comparative evidence facilitates the identification of small ORFs. Second, we identified highly conserved genes of unknown function (e.g., genes 31, 31.1, 31.2). Such a high level of conservation suggests that these gene products are required for phage survival independent of the growth environment; figuring out the function of these gene products should be useful to understanding the SPP1 life cycle. In addition, we can identify gene or gene domains in the structural genes that are under strong purifying selection and thus show no or few mutations versus those that are under positive selection and indicative of adaptation to shifting host and/or environmental conditions. These comparisons can also allow us to identify genes that work together (e.g., the major capsid and the portal) and are thus more functionally constrained in their evolution. Finally, the smaller genome with fewer genes of SPP1 suggests that the process of domestication, i.e., adaptation to lab conditions, may be accompanied by gene loss, analogous to what happens to bacteria species [62,63].

Like the genomes of their hosts, phage genomes evolve by two major mechanisms: mutation (either SNPs or indels) and horizontal gene transfer (either of whole gene or of intragenic modules) [6,53,55,64,65]. Our genome comparisons show evidence of SNPs, insertions/deletions of small genes, as well as intragenic mosaicism. However, in spite of the evidence that these processes are occurring, the gene complements of our three new phages bear remarkable similarity to their distant relative. As is typical of many phages [7,45], the right end of our phage genomes showed less nucleotide conservation and contained more unique genes, many of which cannot be assigned a function. In addition, we noted the splitting/fusion of a gene in our phages as mechanism for genome change. The relative contribution of these various mechanisms to the ability of phages to adapt is still an open question. Some mechanisms, such as mutation, are more likely to be important in newly diverged clades while others occur at lower frequency (e.g., recombination) and will thus be more prevalent in phages that have had more time to diverge.

Most experimental evolution studies have looked at the genetic changes in one phage strain as it interacts with one bacterial strain, precluding opportunities for horizontal gene transfer [66,67], although the authors of [65] have documented rapid genetic changes in genes associated with adsorption to hosts in these pathogens. However, comparative genomic studies of closely related phages clearly show that horizontal gene transfer, particularly of small genes, is an important source of genetic variation in phages [45,53,61]. The above considerations beg the question of what maintains the high level of genetic similarity between phages that are separated by more than 40 years of viral evolution. We suggest that genetic diversity within the host species (e.g., clonal vs. non-clonal) as well as unpredictable host availability may be critical. *Bacillus subtilis* has been shown to have a natural population structure defined by rates of recombination sufficiently high to eliminate strong clonal patterns [68]. Phages that infect hosts whose population genetics reflect high and rapid rates of recombination experience selection that may favor genomes conferring fitness advantages across a range of conditions. One consequence of such selection may be longer term stability of successful gene complements, and the SPP1 phages may exemplify such a pattern.

## Figures and Tables

**Figure 1 viruses-14-02106-f001:**
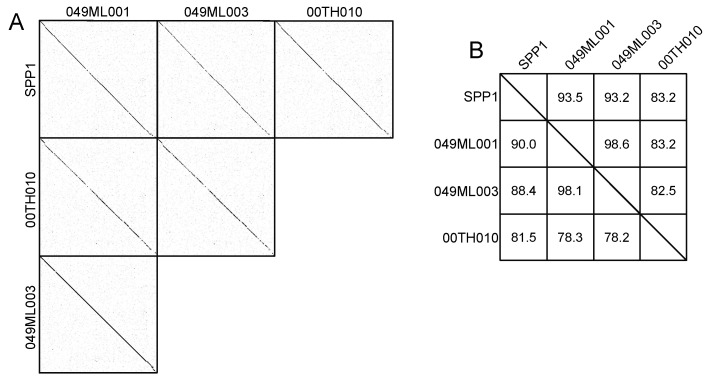
(**A**) Dot plot of the genomes of our three newly isolated SPP1-like phages as well as of SPP1. Plot created using Gepard [49]. (**B**) Average nucleotide identity (ANI; above the diagonal) and average amino acid identity (AAI; below the diagonal) for the SPP1 cluster phages, with each phage compared to the other phages. ANI and AAI were calculated using algorithms from [48].

**Figure 2 viruses-14-02106-f002:**
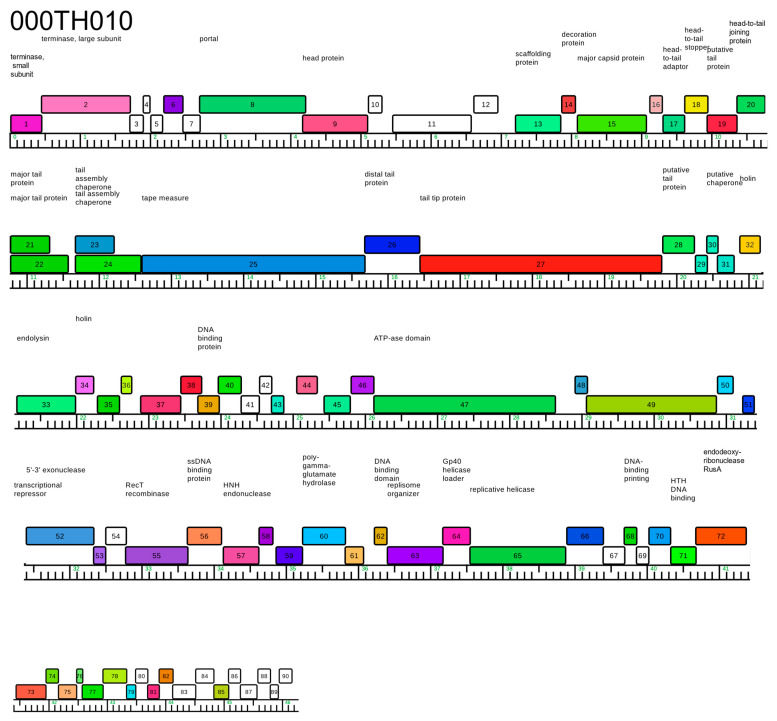
Genome map of 000TH010, chosen because it is the longest genome with the most genes. The ruler shows genome length (in kilobases) with the forward genes shown above the ruler. Function or putative functions are listed above genes (see also Table 2). Map was created using Phamerator [52].

**Figure 3 viruses-14-02106-f003:**
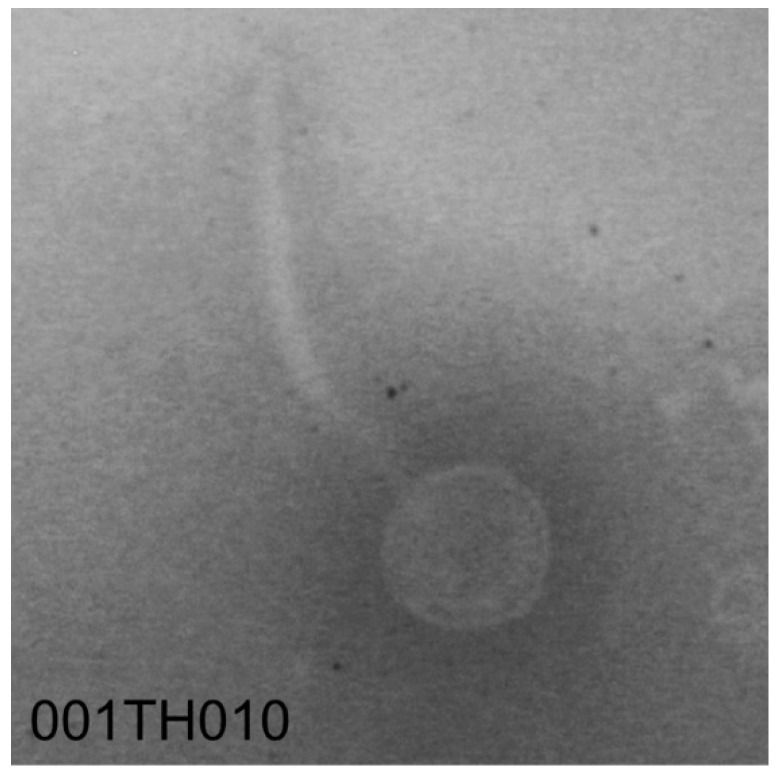
Representative brightfield TEM images of 000TH010 stained with 2% phosphotungstic acid. Images were taken at 50,000× magnification using a Zeiss EM 109.

**Figure 4 viruses-14-02106-f004:**
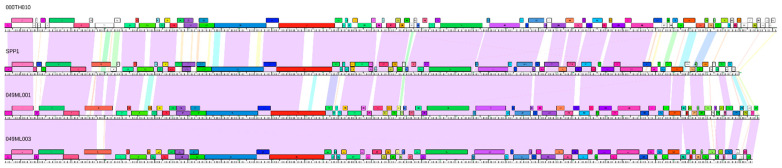
Genome maps of the phages of cluster SPP1. Gene boxes are colored based on pham membership, as determined by amino acid (aa) alignment using BLASTp and Clustalw done within Phamerator [52], using a database of 250 *Bacillus* phages. Genes that are found in only one phage or show limited aa similarity (less than 32.5%) are in white. The color spectrum between two adjacent sequences reflects nucleotide sequence similarity as measured by pairwise local alignment using BLASTn “align two sequences”, with lavender being the most similar (E = 0) and red being the least similar (E = 10^−4^, set as the minimal threshold). A lack of shading indicates that the minimal similarity threshold has not been met (E score greater than 10^−4^). A total of 116 phams among these four phages were identified.

**Figure 5 viruses-14-02106-f005:**
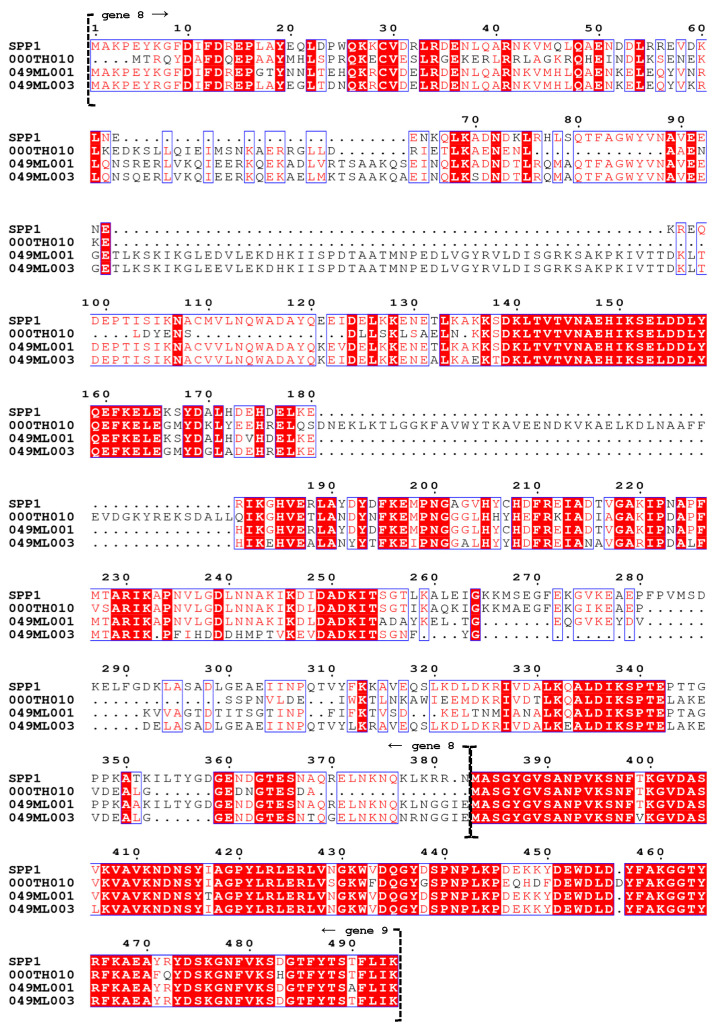
Protein alignment of gene products 8 and 9 in SPP1 and their homologs in our phages. Alignments were constructed using the % Equivalent parameters of ESPript 3.0 [51]. Red lettering indicates sequence and/or physiochemical property matches in the aligned gene products. Black lettering indicates a mismatch in both sequence and physiochemical property of aligned gene products. Red backgrounds indicate identity across all aligned gene products. Blue boxes indicate consensus in both sequence and physiochemical property across the majority of aligned gene products.

**Table 1 viruses-14-02106-t001:** Basic genomic characteristics of three *Bacillus subtilis* phages related to SPP1 as well as SPP1. ORF = Open Reading Frame signifying either known or putative genes. * = information from [24]; † = the two holin genes with an in-frame internal initiation codon were only counted once; N/A = Not available.

Phage	Isolation Strain	Genome Size (bp)	%GC	No of ORFs	Collection Site, Date	GPS Coordinates	Accession Number
**000TH010**	T89-06	46,274	43.8	90	Tumamoc Hill, AZ, September 2011	32°13′10.1″ N, 111°00′12.2″ W	MN176219
**049ML001**	T89-19	45,238	43.7	82	Mount Lemmon, AZ, May 2014	32°20′14.9″ N, 110°41′29.3″ W	MN176227
**049ML003**	T89-19	44,817	43.7	81			MN176228
**SPP1**	168 *	44,016	43.7	79 ^†^	Pavia, Italy 1968 *	N/A	X97918.3

**Table 2 viruses-14-02106-t002:** Comparison of our phages to the revisited annotation of phage SPP1 from [25]. SPP1 gene numbers/names (e.g., 17, 17.1, and 17.1 *) follow naming convention established for SPP1 [25]. We added one putative gene not called previously (gene 46.2) based on homologs present in our other phages. In addition to function listed in [25], we called function for three additional genes (in bold) based on HHPred matches or a published study (†; [26]). If putative homologs were found in our recently isolated phages, % amino acid (aa) alignment/identity with SPP1 are given, as determined by BLASTp.

SPP1 Gene	Function	% AA Alignment/Identity with SPP1		
		**049ML001**	**049ML003**	**000TH010**
1	Terminase, small subunit	100/98.6	100/98.6	95.2/78.4
2	Terminase, large subunit	100/98.6	100/95.3	100/95.5
3		-	-	-
4		100/32.3	100/32.3	94.0/37.0
5		-	-	-
6	Portal protein	100/96.6	100/96.6	94.0/92.2
7	Head protein	100/98.4	100/99.4	100/92.2
8		46.3/76.4	44.5/66.2	98.9/44.1
9		20.8/98.2	20.3/98.2	100/89.4
10		-	-	-
11	Scaffolding protein	100/99.5	100/99.5	100/93.5
12	Decoration protein	100/100	100/100	98.0/82.5
13	Major capsid protein	100/85.3	100/85.3	100/80.5
14		-	-	82.0/91.7
15	Head-to-tail adaptor	100/93.2	100/93.2	99.0/82.2
16	Head-to-tail stopper	100/97.3	100/97.3	100/87.2
16.1	Putative tail protein	100/99.3	100/99.3	100/90.1
17	Head-to-tail joining	100/98.5	100/98.5	97.0/73.5
17.1	Major tail protein, tail tube	100/99.4	100/99.4	100/70.4
17.1 *	Major tail protein, tail tube	100/98.9	100/98.9	100/70.4
17.5	Tail chaperone	96.6/98.8	96.6/98.8	100/64.6
17.5 *	Tail chaperone	98.8/97.0	98.8/97.0	100/60.9
18	Tape measure	100/98.3	100/98.4	100/78.0
19.1	Minor/distal tail protein	100/97.2	100/97.2	100/90.1
21	Tail tip protein	99.9/73.4	99.6/73.5	99.6/84.4
22	Putative tail protein	98.5/87.2	98.5/87.2	100/84.3
23		66.0/55.6	66.0/58.3	100/90.7
23.1	**Putative chaperone ^†^**	95.8/78.3	95.8/78.3	100/94.1
24		92.9/91.3	92.9/91.3	100/88.2
24.1	Component of holin	100/87.1	100/87.1	100/93.5
24.1 *	Component of holin	90.3/85.7	90.3/85.7	90.3/92.9
25	Endolysin	100/98.5	100/98.5	100/93.0
26	Component of holin	100/97.6	100/97.6	100/91.5
26 *	Component of holin	97.6/97.5	97.6/97.5	97.6/91.3
26.1		-	-	91.5/86.1
27		100/92.9	100/92.4	100/87.0
28		100/87.4	100/86.3	98.9/86.2
29	DNA binding protein	100/97.0	100/97.0	100/90.0
29.1		100/98.1	100/98.1	100/85.7
30		55.7/97.4	54.9/97.4	47.0/89.7
30.1		100/98.2	100/98.2	100/85.7
31		100/100	100/100	100/90.4
32	ATP-ase	100/99.2	100/98.9	100/95.1
32.5		100/94.6	100/94.6	100/83.9
33	Putative bacteria surface binding protein	98.3/97.3	98.3/97.5	98.3/94.1
33.1		100/91.7	100/98.6	98.6/84.3
34	Transcriptional repressor	100/96.2	100/96.2	100/94.2
34.1	5′-3′ Exonuclease	100/95.8	100/96.1	100/94.5
34.2		100/50.8	100/50.8	100/44.3
34.3		100/95.2	100/96.4	-
34.4		100/70.1	100/70.1	-
35	RecT-like recombinase	100/99.3	100/99.3	100/94.8
36	ssDNA binding protein	100/92.5	100/92.5	100/74.8
36.1	HNH endonuclease	100/98.2	100/98.2	100/96.9
37		100/49.6	100/50.4	100/88.4
37.1	Poly-gamma-glutamate hydrolase	100/97.0	100/97.0	100/87.8
37.2		100/96.4	100/96.4	100/84.5
37.3	DNA binding domain	100/70.2	100/70.2	100/61.4
38	Replisome organizer (PriA-like)	100/96.5	100/98.0	99.6/91.4
39	Gp40 helicase loader	100/95.2	100/95.2	100/91.3
40	Replicative DNA helicase	100/98.4	100/98.4	100/97.5
41		100/88.6	100/88.6	100/55.0
42		87.8/85.2	87.8/85.2	-
42.1	**DNA-binding protein**	-	96.4/94.4	91.2/59.6
42.2		100/98.0	100/98.0	98.0/87.6
43	**HTH DNA binding**	100/90.6	100/90.6	99.1/61.7
44	Endodeoxyribonuclease RusA	88.0/92.3	88.0/92.3	87.0/78.3
46		-	-	-
46.1		100/98.1	100/98.1	100/90.3
46.2		100/100	100/100	97.1/97.0
47		100/92.4	100/91.6	96.6/68.7
48		-	-	-
49		-	-	-
50		-	-	-
50.1		-	-	-
51		100/97.0	-	-
51.1		-	-	-
52		100/50.7	100/48.8	-
53		100/95.6	100/63.2	100/63.2

## Data Availability

All genome sequences have been deposited in the GenBank database of NCBI and accession numbers are listed in Table 1.

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
