# Peer review of "Forty Years without Family: Three Novel Bacteriophages with High Similarity to SPP1 Reveal Decades of Evolutionary Stasis since the Isolation of Their Famous Relative"

_viruses, 2022, doi:10.3390/v14102106_

Round 1

Reviewer 1 Report

The focus of this study was comparative genomics of three newly sequenced bacteriophages compared to a classically studied Bacillus subtilis phage SPP1. The authors use appropriate methods and approach to complete this work. They are able to provide some comparative genomics that was previously not available because of their new sequences, and they provide updates to the SPP1 annotation. The comments below are to improve the manuscript, with a particular focus on what may be some inconsistencies the coloring of protein phams in Figure 3.

It would be interesting to comment on host range, if this has been tested, since the authors state that the B. subtilis strain they used is genetically and physiologically distinct from the canonical B. subtilis 168 and other strains in their collection. I don’t recommending extra wet lab work to this genomics paper, and imagine this might be part of a future, large comparative study. But if it exists already, context of phage growth on the well known B. subtilis 168 would be nice to include.

Do TEM images of these phages exist? This would also be nice to add.

Figure 1: Consider adding some vertical and horizontal lines to demarcate individual genomes. It would make it easier to see SPP1/ML001/ML003 are most similar, in a glance.

Line 232, not sure I agree with the word “crucial” to describe SPP1 gp9 given the unknowns about essentiality, function and translation. Are there any hints from coding potential? Do gp8/9 reside in different reading frames?

Line 243, ‘and but also’ has an extra word, please update for grammar

Figure 3: We do not usually see novel genes (that appear white in Phamerator) in regions where pairwise shading is lavender (E value = 0, but also usually accompanied with a very high percent identity. It would be good to know if this is a Phamerator glitch or real sequence differences. There are several regions in the comparative genome map that may need pham color updating or clarify in the text that these are actually novel genes please.

-SPP1 genes 24.1/24.1* and homologs appear as white rectangles, yet retain purple shading between genomes. Those are not novel phams, correct? Table 3 indicates >85% amino acid identity between these proteins.

-OOTH001 gp10 is annotated and doesn’t appear in any other genomes, yet retains purple shading between genomes. Do you see coding potential in this region, in the other genomes?  I can’t compare this one to Table 3 since the Table is SPP1 focused. I poked around with a quick/rough tblastx and you might have a more distant relationship here (60% identity), but please check with the proper sequences and tools to see if there really may be ORFs. 

-SPP1 gp14 vs 000TH010 gp16 retain purple shading between genomes. This ORF should have an entry in Table 3, 000TH010 column, and doesn’t.

-ML001/ML003 gp34and gp 35 retain purple shading between genomes. This isn’t in Table 3 since the Table is SPP1 focused, and I didn’t check this amino acid identity myself. Please take a look?

Once those are clarified, it might be nice to check if the genes that appear novel (white) in your phamerator map also lack homologs in genbank? If so, or not, it might give you some additional perspective on 40 years of ‘evolutionary stasis’ as well as horizontal gene transfer.

One of the most interesting implications covered in this paper is finding very similar bacteriophages using different host bacteria, 40 years apart. I would like to see this Kimberly Seed has been isolating V. cholerae bacteriophages over decades (https://journals.asm.org/doi/10.1128/mbio.03088-21) , and finds some interesting stepwise variability in their genome sequences. You don’t really need more references but it’s uncommon to see such temporal analysis and if you haven’t looked at this work it might give you some interesting perspective.
